# Mixture of Contexts for Long Video Generation

**Shengqu Cai**[1,2]    **Ceyuan Yang**[2]*   **Lvmin Zhang**[1]    **Yuwei Guo**[2,4]    **Junfei Xiao**[2,3]
**Ziyan Yang**[2]    **Yinghao Xu**[1]    **Zhenheng Yang**[5]    **Alan Yuille**[3]    **Leonidas Guibas**[1]
**Maneesh Agrawala**[1]    **Lu Jiang**[2]    **Gordon Wetzstein**[1]
[1]Stanford University    [2]ByteDance Seed    [3]Johns Hopkins University
[4]CUHK    [5]ByteDance

## ABSTRACT

Long-context video generation is fundamentally a memory problem: models must retain and retrieve salient events across long range without collapsing or drifting. However, scaling diffusion transformers (DiTs) to generate long-context videos is fundamentally limited by the quadratic cost of self-attention, which makes memory and computation intractable and difficult to optimize for long sequences. We recast long-context video generation as an internal information retrieval task and propose a simple, learnable sparse attention routing module, Mixture of Contexts (MoC), as an effective long-term memory retrieval engine. In MoC, each query dynamically selects a few informative chunks plus mandatory anchors (caption, local windows) to attend to, with causal routing that prevents loop closures. As we scale the data and gradually sparsify the routing, the model allocates compute to salient history, preserving identities, actions, and scenes over minutes of content. Efficiency follows as a byproduct of retrieval (near-linear scaling), which enables practical training and synthesis, and the emergence of memory and consistency at the scale of minutes. Project Page: `https://primecai.github.io/moc/`.

## 1 INTRODUCTION

Video generation has emerged as a central problem in generative modeling, powering content creation, simulation for autonomous systems, and interactive storytelling. Recent Transformer-based diffusion models can synthesize increasingly realistic clips by modeling complex space–time dependencies; yet, pushing them to minute- or hour-long horizons exposes a deeper challenge: long-term memory. Models must retain and retrieve salient events across extended timelines without drift, collapse, or loss of identity. Dense self-attention becomes computationally prohibitive as sequences grow, and moreover, the core difficulty is not merely computational, but learning to selectively recall the right context at the right time.

A salient characteristic of video data is its high degree of temporal redundancy: consecutive frames frequently exhibit much pixel similarity or only minor motion, resulting in substantial repetition of information across the sequence. Therefore, prior efforts reduce cost either by compressing history into compact representations (e.g., keyframes (Henschel et al., 2025; Xiao et al., 2025a), frame packs (Zhang & Agrawala, 2025), and latent states (Dalal et al., 2025; Po et al., 2025)), or by imposing fixed sparse or selective patterns that thin interactions across the sequence (Li et al., 2025b; Zhang et al., 2025c; Xi et al., 2025; Xiao et al., 2025b; Yu et al., 2025a). These strategies lengthen the feasible horizon but hard-code a compromise between efficiency and fidelity: compressed summaries lose detail, and static sparsity or selection cannot adapt to which past events matter at each step, thereby limiting the preservation of long-range dependencies and narrative coherence.

In this work, we reformulate long-context video generation as an internal information retrieval process, where each token dynamically accesses only the most relevant context through learnable sparse attention routing. To realize this, we propose an adaptive Mixture of Contexts (MoC) framework that learns to route each query to the most relevant segments of the video sequence, instead of relying

---

*Corresponding author.

on uniform or static sparse attention or a fixed selection strategy. Specifically, MoC partitions the multi-modal token stream into content-aligned chunks along frames, shots, and captions, then lets each query select only a few relevant chunks via a parameter-free yet trainable top-$k$ router. Two mandatory anchors: cross-modal links to all text tokens and intra-shot local window links are activated to stabilize local fidelity while reserving routing capacity for genuinely long-range recall. A causal routing mask is additionally applied to prevent pathological loop closures by enforcing a directed acyclic interaction graph, improving roll-out robustness over minute-scale sequences. For efficient implementation, the selected key tokens are directly processed by the flash-attention kernel, which supports variable sequence lengths and high throughput. During training, we progressively adjust the granularity of chunks and the selectivity of the routing mechanism, resulting in a gradual sparsification that encourages the model to focus on the most informative context as training progresses.

We show that replacing dense self-attention with our Adaptive Mixture of Contexts (MoC) reframes long-video generation as internal in-context retrieval. A learned sparse context routing policy allocates compute to salient history and sustains cross-shot identities, actions, and layouts over minutes-long sequences, without modifying the diffusion backbone or its training recipe. Efficiency follows as an enabler, as MoC prunes over 85% of token pairs and reduces the attention FLOPs budget by up to $7\times$, yielding a measured $2.2\times$ end-to-end generation speedup on minute-scale scenes ($\approx$180k tokens). In short, our MoC is the first work that demonstrates learned sparse context routing could overcome the practical barriers of quadratic attention, and effectively deliver minutes-level long-context video memory at near short-video cost, while maintaining and often surpassing the fidelity and consistency of dense baselines.

## 2 RELATED WORK

The prohibitive $O(L^2)$ computational cost of standard self-attention mechanisms in Transformer architectures (Vaswani et al., 2017; Peebles & Xie, 2023) becomes the primary obstacle when applied to the vast sequence lengths involved, and the difficulty of maintaining coherence and preventing visual degradation over long time horizons. Our work builds upon prior efforts in efficient sequence modeling and long-video generation frameworks.

**Long Video Generation.** Existing video generation models (Kong et al., 2024; Team, 2024; Guo et al., 2024; Yang et al., 2024b; HaCohen et al., 2024; Bar-Tal et al., 2024; Hong et al., 2023; Chen et al., 2023; 2024; Ge et al., 2023; Zhang et al., 2023) are mostly limited to a few seconds. To push beyond this short horizon, TECO (Yan et al., 2023) introduces temporally consistent transformers with a recurrent state to propagate information over long sequences, and NUWA-XL (Yin et al., 2023) adopts a diffusion-over-diffusion hierarchy that generates extremely long videos by first synthesizing sparse keyframes and then recursively filling in between. Several recent frameworks specifically target longer video generation using autoregressive models that operate on frames, chunks, or segments, such as MALT (Yu et al., 2025b) and CausVid (Yin et al., 2025). While these frameworks extend generation capabilities, they often grapple with error accumulation (Wang et al., 2025) inherent in sequential prediction or face uncertain computational scaling to longer durations. To mitigate these issues, RollingDiffusion (Ruhe et al., 2024) and Diffusion Forcing (Chen et al., 2025a) inject controlled noise into the historical context and train the model to denoise it, increasing robustness to compounding errors. MAGI-1 (Sand-AI, 2025) and SkyReels-V2 (Chen et al., 2025b) scale up these ideas by employing autoregressive denoising, aiming for potentially longer durations. An orthogonal strategy is to distill the entire past into a constant-size latent. TTTVideo (Dalal et al., 2025) and LaCT(Zhang et al., 2025f) use a learnable MLP to encode the context during inference, while FramePack (Zhang & Agrawala, 2025) encodes arbitrarily many frames into a fixed vector for next-frame prediction. FramePack (Zhang & Agrawala, 2025) also proposes early planning of future frames to mitigate the error accumulation issue. This is similar to using keyframes or anchor frames (Henschel et al., 2025; Weng et al., 2023; Long et al., 2024; Zhao et al., 2025; Hu et al., 2025; Xie et al., 2024; Yang et al., 2024a; Xiao et al., 2025a), where certain frames are predefined and the video generation model only does an interpolation sampling job. These methods extend video generation to the one-minute range but still face a hard ceiling on maintaining long-context coherence going forward, as they rely on lossy compression of the contexts. The work most closely related to ours is Long-Context Tuning (Guo et al., 2025) (LCT), which starts from a single-shot DiT and expands its context window to a scene comprising up to eight shots ($\approx$8s, $\sim 2.3\times10^4$ tokens each). LCT (Guo et al., 2025) keeps the attention mechanism dense: all text and video tokens inside the

enlarged window attend to one another after being positioned with an interleaved 3D RoPE. While this design elegantly re-uses the pretrained weights and yields impressive multi-shot coherence, it inherits the quadratic cost of full self-attention – FLOPs and memory scale with $(8L_{\text{shot}})^2$.

**Sparse Attention for Video Generation.** Sparse attention leverages the observation that attention matrices are often sparse (many scores are near zero) and computes attention only for a subset of important token pairs, a natural fit for video generation given spatiotemporal redundancy. Training-free pruners include SparseVideoGen (Xi et al., 2025), which profiles heads that dynamically specialize into spatial vs. temporal and selects a per-head pattern, and STA (Zhang et al., 2025e), which exploits localized 3D windows by operating tile-by-tile over FlashAttention-friendly blocks (Dao et al., 2022; Dao, 2024). Universal filters such as SpargeAttn/SageAttention (Zhang et al., 2025c;b;a) combine selective token compression with a softmax-aware pass to skip parts of $QK^T/PV$, and AdaSpa (Xia et al., 2025) proposes a "blockified" dynamic pattern with Fused LSE-Cached Search that reuses sparse indices across denoising steps. Jenga (Zhang et al., 2025g) uses training-free block-wise attention carving plus progressive resolution. Beyond these post-hoc pruners, recent trainable or structured designs include VMoBA (Wu et al., 2025), which learns a mixture-of-block scheme with layer-wise partitions and global/thresholded block selection for VDMs. VSA (Zhang et al., 2025d) proposes a hardware-efficient coarse-to-fine sparse kernel that replaces full attention at both training and inference. Radial Attention (Li et al., 2025b) instead, uses a static $\mathcal{O}(n \log n)$ mask derived from spatiotemporal energy-decay that enables longer generations with near-dense quality. While these advances substantially reduce costs and accelerate video generation, most methods either prune emergent dense maps or impose fixed sparsity priors, focusing on accelerating the generation of short videos. By contrast, our Mixture of Contexts learns deliberate, end-to-end routing of context sources and focuses on long context memory/consistency, with acceleration as a byproduct of sparsity.

**Context Learning in Visual Generation.** A complementary line of work treats context—past frames, states, or reference images—as a first-class signal for learning and control. For video world models, where action and camera position signals are available, WORLDMEM (Xiao et al., 2025b) augments simulators with an external memory bank of frames and states and retrieves relevant entries via Field-of-View (FoV) overlapping to preserve long-term scene consistency. A similar work, Context-as-Memory (Yu et al., 2025a), targets interactive long videos, explicitly retrieving a small set of historical frames as conditions for each step to sustain scene consistency, also via FoV overlapping to select the relevant frames. Concurrently, VMem (Li et al., 2025a) uses a surfel-indexed, occlusion-aware memory to retrieve relevant views and maintain consistency under re-visits. Back to the image space, IC-LoRA (Huang et al., 2024a) demonstrates that DiTs already exhibit in-context abilities and proposes concatenating reference images with lightweight task-specific LoRA (Hu et al., 2022) to adapt across tasks with few samples. DSD (Cai et al., 2025) turns in-context generation into paired supervision via self-distillation: curate image grids with a VLM, then fine-tune a text+image-to-image model. OminiControl (Tan et al., 2025) offers a parameter-efficient, unified framework for image-conditioned control in DiTs, enabling broad conditioned tasks without auxiliary modules. Recent open-sourced models, such as FLUX-Context (Labs et al., 2025), concatenate text and images to unify in-context image generation and editing, with improved consistency. These works demonstrate that, given a sufficiently large training scale, routing and in-context learning are very powerful in extracting useful information from contexts. Our Mixture of Contexts follows this routine, and proposes to learn to route among multiple context sources end-to-end, enabling deliberate selection and composition of contextual signals rather than relying solely on fixed retrieval or a single conditioning pathway.

## 3 METHOD

To generate long videos without incurring the quadratic cost of standard self-attention, our method replaces the DiT (Peebles & Xie, 2023) backbone's dense attention with an adaptive, content-aligned Mixture of Contexts (MoC) layer. At a high level, MoC (i) routes each query only to the most relevant chunks of context, (ii) aligns those chunks with natural video boundaries such as frames, shots, and caption tokens, and (iii) enforces causality so information flows strictly forward in time. The following subsections detail the routing formulation (Sec. 3.1), chunking and selection strategy for the interleave text-to-video generation (Sec. 3.2), computation efficiency (Sec. 3.3). The overall pipeline of our method is shown in Fig. 1.

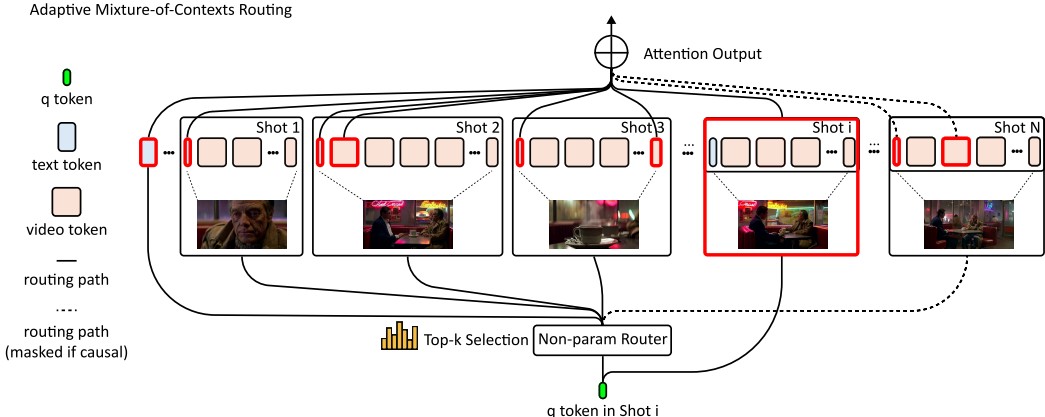

Figure 1: **Overview of our Adaptive Mixture of Contexts.** Given a long multi-modal token stream, we first tag natural boundaries (frames, shots, text segments) and slice the sequence into content-aligned chunks (blue and pink blocks for texts and videos, respectively). Each chunk's keys are then mean-pooled to obtain a single representative vector. For every query token $q$ (green), we compute the dot-product between $q$ and every pooled key, apply a top-$k$ operation, and add mandatory links (global caption and intra-shot edges). The result fetches only a selected subset of chunks, which are forwarded to Flash-Attention – while all other tokens are skipped, yielding near-linear compute and memory in the number of retrieved chunks rather than quadratic in total sequence length.

## 3.1 MIXTURE OF CONTEXTS

**Vanilla Attention in Diffusion Transformers.** We first revisit the attention module commonly used in Diffusion Transformers (DiT) (Vaswani et al., 2017; Peebles & Xie, 2023), the backbone of state-of-the-art video generation models. An attention module is defined as:

$$\text{Attn}(\boldsymbol{Q}, \boldsymbol{K}, \boldsymbol{V}) = \text{Softmax}\left(\frac{\boldsymbol{Q}\boldsymbol{K}^\top}{\sqrt{d}}\right) \cdot \boldsymbol{V}, \tag{1}$$

where $\boldsymbol{Q}$, $\boldsymbol{K}$, and $\boldsymbol{V}$ denote the query, key, and value features, respectively, while $d$ stands for the feature dimension. Note that when we consider $\boldsymbol{Q} = \{\boldsymbol{q}_i\}$ as a set of independent vectors, Eq. 1 can be written as $\text{Attn}(\boldsymbol{q}_i, \boldsymbol{K}, \boldsymbol{V}) = \text{Softmax}(\boldsymbol{q}_i \boldsymbol{K}^\top / \sqrt{d}) \cdot \boldsymbol{V}$ that performs in query-wise.

**Dynamic Routing via Top-$k$ Selection.** In a video DiT (Peebles & Xie, 2023), the sequence length easily scales up to nearly 200k for a 480p, 1-minute-long video. This makes the $O(L^2)$ computation of self-attention extremely expensive. Due to feature redundancy, a common practice is to divide the video sequence into several chunks, allowing a query token to interact with only a subset of these chunks. Autoregressive video generation works (Yin et al., 2025; Chen et al., 2025b;a) often split context by frames as chunks, where the query $\boldsymbol{q}_i$ attends only to the closest few chunks, losing context beyond a limited distance. Instead, we adopt a learned routing strategy, where each $\boldsymbol{q}_i$ is routed to the most relevant chunks with

$$\text{Attn}(\boldsymbol{q}_i, \boldsymbol{K}, \boldsymbol{V}) = \text{Softmax}\left(\frac{\boldsymbol{q}_i \boldsymbol{K}_{\Omega(\boldsymbol{q}_i)}^\top}{\sqrt{d}}\right) \cdot \boldsymbol{V}_{\Omega(\boldsymbol{q}_i)}, \tag{2}$$

where $\Omega(\cdot)$ yields a set of routed indices, and $\Omega(\boldsymbol{q}_i)$ is the indices of all interested context positions for the query $\boldsymbol{q}_i$. Given the list of all chunks $\Phi$, for every $\boldsymbol{q}_i$, only a few chunks are considered for attention computation with a top-$k$ operation

$$\Omega(\boldsymbol{q}_i) = \left[\arg\max_{\Omega^*} \sum_{\omega \in \Omega^*} \left(\boldsymbol{q}_i^\top \phi(\boldsymbol{K}_\omega)\right)\right] \qquad \text{where} \quad \Omega^* \subseteq \Phi \text{ and } |\Omega^*| = k, \tag{3}$$

where $[\cdot]$ concatenate and join all indices of the top-$k$ chunks. The relevance between the $\boldsymbol{q}_i$ and the chunk sequence $\boldsymbol{K}_\omega$ is determined by the inner product of $\boldsymbol{q}_i$ and the descriptor for $\boldsymbol{K}_\omega$ denoted

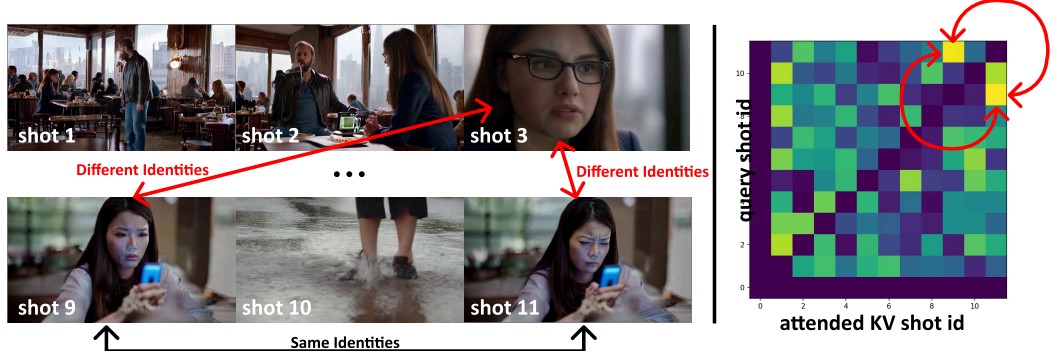

Figure 2: **Illustration of loop closures without causality.** *Left:* successive frames from an ablation model without causal masking. After a café scene (top row), the story is meant to cut to a riverbank shot of the same woman looking at her phone (bottom row). However, because shot 9 strongly routes to shot 11 while shot 11 simultaneously routes back to shot 9, the model becomes trapped in a two-node feedback loop, so that shot 9 and 11 have limited communication with earlier shots, as shown in the routing counts (*right*).

as $\phi(\boldsymbol{K}_\omega)$. For this work, we use the simple, efficient, yet effective mean pooling operation as the descriptor transformation $\phi$. We argue that such a mean pooling operation is highly sufficient and expressive for video generation tasks. The effectiveness of this design relies on the intrinsic quality of the Diffusion Transformer's learned features. As demonstrated by DDAE (Xiang et al., 2023), denoising diffusion autoencoders function as unified self-supervised learners, naturally acquiring semantically meaningful and linearly separable internal representations. Consequently, the global average of a token chunk effectively captures its dominant semantic content and visual layout, providing a robust summary that enables queries to distinguish relevant context based on high-level alignment. This approach effectively captures dominant semantic features while being robust to local variations, a property that translates naturally to video chunks where spatially and temporally adjacent tokens often represent redundant or correlated visual elements (e.g., static backgrounds or gradual motions). Furthermore, in our trainable framework, this pooling is not a static heuristic but an adaptive mechanism: while top-$k$ itself is non-differentiable, the model learns indirectly through the attention mechanism on selected chunks. Specifically, if a selected chunk proves irrelevant during attention computation, gradients from the loss will flow back through its keys/values, which is the source of the mean-pooled descriptor. This process attenuates unhelpful representations and encourages the query/key projections to produce more discriminative similarities over training iterations. This self-correcting process aligns with indirect adaptation seen in hard-routing MoE systems and sparse attention frameworks (e.g., where downstream modules provide the learning signal despite discrete and non-differentiable selections). This end-to-end differentiability and parameter-less router ensures that the seemingly simple dot-product routing becomes highly expressive, as the network shapes embeddings to emphasize discriminative features for sparse attention, without introducing additional parameters or computational overhead. Empirical zero-shot application to pretrained models further validates its efficacy, as will be detailed in our supplementary material.

**Context Drop-off.** To enhance the robustness of our Mixture of Contexts (MoC) and mitigate issues akin to the "dead expert" problem in Mixture-of-Experts (MoE) systems, we first introduce *context drop-off*. Motivated by the observation that routing may suffer from inaccuracies due to noise in embeddings or evolving data distributions, this technique randomly removes a subset of the top-$k$ selected chunks for each query token. Specifically, for a given query $q_i$, after computing the routed indices $\Omega(q_i)$ in Eq. 3, we sample a drop probability $p_{\mathrm{drop}} \sim \mathrm{Uniform}(0, p_{\max})$ and mask out $\lfloor p_{\mathrm{drop}} \cdot k \rfloor$ randomly chosen chunks from $\Omega(q_i)$. This forces the model to generate coherent outputs even when a certain chosen context is sporadically unavailable, promoting redundancy in the learned dependencies and preventing catastrophic failure from routing errors.

**Context Drop-in.** Complementarily, we employ *context drop-in* to inject extraneous chunks into the selected set to simulate over-inclusive routing. For each query, we randomly sample $m \sim \mathrm{Poisson}(\lambda)$

chunks to be included in the selected pool $\Omega(q_i)$. This technique combats the dead route problem by artificially activating underutilized chunks, ensuring gradients flow through a broader range of context segments and balancing the routing distribution over time. Since our router is parameter-less and relies solely on mean-pooled feature similarity, these regularization techniques do not interfere with the learning of the routing mechanism itself. Instead, if a chunk is truly important, its relevance will be naturally enhanced through backpropagation in the attention modules, as the model adjusts the query and key projections to amplify meaningful similarities. In essence, the end-to-end differentiability of the system means that the attention process implicitly serves as the router's learning signal, making the framework self-correcting and adaptive without dedicated routing parameters.

**Per-Head Distributed Routing.** A crucial design choice in Mixture of Contexts is the granularity of the retrieval process: is context selected globally once, or dynamically at every step? We implement routing at the finest granularity – independently for each attention head in every layer. Rather than relying on a single "global" router to select a fixed set of $k$ chunks shared across the entire network, our approach effectively acts as an ensemble of $L_{\text{layers}} \times H_{\text{heads}}$ independent routers. This distinction is vital for two reasons. First, different attention heads in diffusion transformers specialize in distinct feature subspaces (e.g., low-level texture coherence vs. high-level semantic identity), necessitating access to different historical segments. Second, while each head is strictly sparse (attending to only $k$ chunks), the union of selected chunks across all heads and layers covers a significantly larger portion of the context. This distributed routing ensures sufficient global communication and prevents the information bottleneck that would arise from a static global selection, allowing the model to utilize its entire parameter space to reconstruct the full context manifold through diverse, sparse viewpoints.

## 3.2 ATTENTION CHUNKING AND ROUTING

**Content-aligned Chunking.** A critical and often overlooked design axis in Mixture of Contexts is how we carve the gigantic token stream into candidate chunks. In long-context LLMs this decision is trivial: the input is a homogeneous 1D sequence of sub-word tokens endowed with a single RoPE (Su et al., 2021), so slicing it into fixed-length windows, such as in MoBA (Lu et al., 2025), both preserves local semantic coherence and matches the monotone positional metric. Video generation DiTs (Peebles & Xie, 2023), by contrast, are often multi-modal, and operate on a heterogeneous 3D+modality lattice: a flattened order that interleaves spatial patches, temporal frames, text tokens, which have separate 3D RoPE (Su et al., 2021) factors. Two neighboring indices may therefore lie far apart in space-time or span an abrupt shot cut, while a static background patch can repeat for hundreds of frames next to a single highly entropic motion token. Uniform windows blur these disparate signals, polluting the mean-pooled key used in Eq. 3 and forcing the top-$k$ selector to waste slots on keys that are internally inconsistent. We instead partition the sequence along content-aware boundaries—frames, shots, and modality stripes – so that each chunk is semantically homogeneous and geometrically local in the 3D positional manifold. This alignment preserves the discriminative power of Eq. 3's mean-pooled keys, yields more informative top-$k$ retrieval, and slashes quadratic overhead without sacrificing long-range coherence. Such a chunking strategy can not only deal with existing single-shot text-to-video generators, but also is compatible with the existing long-video generation approach (Guo et al., 2025), which directly computes attention on an extremely long sequence with interleaved text-video pairs.

**Fixed Cross-Modal Selection as Attention Sink.** In addition to dynamically routed visual chunks, we explicitly require every visual query token to attend to all text tokens in the sequence. This design mirrors a naive use of "sink tokens (Xiao et al., 2024)": a small, persistent set of tokens that every query can attend to, which (i) provides a low-entropy, semantically meaningful anchor for the attention distribution, (ii) guarantees at least one well-conditioned dense block in each attention matrix, and (iii) creates a global gradient highway. We directly use the text tokens, as they typically constitute less than 1% of all tokens, while encoding the most semantically informative signals—specifying global style, character identities, and key actions. The computational overhead is negligible, yet the benefits are substantial: anchoring generation to the prompt significantly reduces prompt-drift errors and prevents the fading of rare attribute words during long video roll-outs. Furthermore, this hard cross-modal link facilitates joint gradient propagation into both text and visual embeddings, tightening their shared latent space and markedly improving editability in downstream tasks such as text-guided video editing.

**Fixed Intra-Shot Selection as Local Window.** Long videos naturally exhibit a strict hierarchical structure, with frames nested within shots and shots within scenes. To leverage this, we explicitly

enforce the intra-shot connections in the attention mechanism, ensuring that each token always attends to its belonging shots—capturing object trajectories, lighting continuity, and other predictive cues. This design allows the Mixture of Contexts (MoC) framework to allocate its sparse attention budget to genuinely long-range dependencies, rather than redundantly modeling local context. Enforcing such connections offers several benefits: it guards against semantic discontinuities at scene cuts where adjacent tokens may become unrelated; it guarantees that every attention matrix contains at least one well-conditioned block; and it provides a contiguous, memory-efficient fallback path even under aggressive adaptive pruning. This strategy is particularly effective when fine-tuning pretrained video generation models, as it preserves the fidelity of each shot from the outset and enables the model to gradually learn to align broader contextual information during training.

**Causality in Sparse MoC.** Sparse routing inherently introduces directionality into the token interaction graph, as each chunk selects a limited set of other chunks for attention. However, in the absence of explicit ordering constraints, this process can degenerate into pathologically closed loops. For example, in ablation studies where each chunk was permitted to select only a single peer, we frequently observed cases where chunk 5 routed to chunk 6 while chunk 6 simultaneously routed back to chunk 5, forming an isolated two-node cycle (see Fig. 2). Such self-loops localize information, obstruct gradient propagation, and manifest as stalled motion or repeated frames during bidirectional generation. To address this, we impose a causal mask at the routing stage, restricting each chunk to attend only to keys from earlier positions in the sequence; specifically, any edge $(i \rightarrow j)$ with $j \geq i$ is masked out prior to top-$k$ selection. This constraint transforms the routing graph into a directed acyclic graph (DAG), ensuring that information flows strictly forward in time and structurally precluding closed cycles. Empirically, causal routing not only eliminates isolated feedback pairs but also promotes richer long-range dependencies, resulting in smoother temporal dynamics and more stable training.

## 3.3 COMPUTATION EFFICIENCY

**Combination with Flash-Attention Kernels.** Dealing with content-aligned and highly unequal chunk sizes is substantially more complex than the evenly split setting, such as in MoBA (Lu et al., 2025) and NSA (Yuan et al., 2025). To accommodate frame, shot, and modality structure while preserving efficiency, we implement an adaptive attention mechanism that operates entirely on GPU, while explicitly exploiting the structural cues in video DiTs (Peebles & Xie, 2023). We first tag the flattened token stream with frame, shot, and caption boundaries and use `torch.bucketize` and prefix-sum tables (`cu_seqlen`, `cu_shot`, etc.) to derive content-aligned, variable-length chunks whose start and end indices coincide with those boundaries, ensuring that each chunk is semantically homogeneous. Boundary information is also used to build a pre-routing mask: forced links (e.g., caption–visual, intra-shot self edges) are inserted before the top-$k$ sparsification step, guaranteeing that the router never spends budget on a chunk that is already mandatory. For each surviving chunk, we obtain a single representative key by on-the-fly `segment_reduce` mean pooling, thus avoiding materializing whole chunks and keeping memory flat even when chunk sizes differ by orders of magnitude. Tokens are gathered in head-major order (via `rearrange(..., 's x h d → h s x d')`) so that the ensuing gathers are coalesced, and the heterogeneous (query, key) pairs are packed into a single Flash-Attention (Dao et al., 2022; Dao, 2024) var-len call. This design yields an attention kernel that respects video-specific constraints while remaining memory- and compute-efficient across millions of tokens. Since all operations involved are head-independent, we can fully utilize tensor parallelization and sharding computations across devices.

**Saved FLOPs.** For each attention head, let $L$ be the sequence length or number of query tokens, $C$ be the number of content-aligned chunks, $k$ be the top-$k$ chunks a query token keeps, $\bar{m}$ be the average length of those selected chunks, and $d$ be the head dimension. Mean-pooling keys inside each chunk costs only $Ld$ adds and is negligible. Routing then evaluates one inner product per query–chunk pair, costing $2LCd$ FLOPs ($\times 2$ since an inner product is one multiplication + one addition per dimension). Finally, fine-grain attention on the pruned set performs $\boldsymbol{QK}$ and $\boldsymbol{PV}$ products over at most $k\bar{m}$ keys per query token, for roughly $4Lk\bar{m}d$ FLOPs. Summing the three terms yields:

$$\text{FLOPs}_{\text{MoC}} \approx Ld + 2LCd + 4Lk\bar{m}d. \tag{4}$$

For the same $L$ and $d$, a vanilla full attention head costs

$$\text{FLOPs}_{\text{dense}} = 4L^2d. \tag{5}$$

| Method | Subject Consistency ↑ | Background Consistency ↑ | Motion Smoothness ↑ | Dynamic Degree ↑ | Aesthetic Quality ↑ | Image Quality ↑ | Sparsity ↑ | FLOPs ↓ |
|---|---|---|---|---|---|---|---|---|
| LCT (Guo et al., 2025) | 0.9378 | 0.9526 | 0.9859 | 0.4583 | 0.5436 | **0.5140** | 0% | $1.7 \times 10^{13}$ |
| **Ours** | **0.9421** | **0.9535** | **0.9920** | **0.5625** | **0.5454** | 0.5003 | **85%** | $\mathbf{2.3 \times 10^{12}}$ |

Table 1: **Multi-shot video generation quantitative comparison.** Under an 85% sparsity, our method reduced FLOPs by **>7×**, while the overall performances often improved.

Their ratio then simplifies to:

$$\frac{\text{FLOPs}_{\text{dense}}}{\text{FLOPs}_{\text{MoC}}} \approx \frac{2L}{Cd + 2k\bar{m}}, \tag{6}$$

which grows linearly with sequence length. For example, given a popular compression ratio of VAE ($16\times$ spatial and $4\times$ temporal downsampling rate), a video with a resolution of 480P, 12fps, and a 1-minute duration becomes a sequence with around 180k tokens. Supposing we use $\bar{m} \approx 1024$, $k = 5$, $C = 36$, $d = 128$, we can calculate that $\text{FLOPs}_{\text{MoC}} \approx 2.32 \times 10^{12}$, while in comparison, dense self-attention on the same sequence costs $\text{FLOPs}_{\text{dense}} \approx 1.66 \times 10^{13}$, hence the adaptive Mixture of Contexts layer reduces multiply–adds by a factor of $> \mathbf{7\times}$.

## 4 EXPERIMENTS

We conduct our main experiment on long scene-level text-to-video generation with multiple shot cuts, a significant use case in AIGC video generation.

**Base Model.** We build our model on a long-context video generator, LCT (Guo et al., 2025), which is the only available architecture that supports long, multi-shot video generation for general scenes. LCT adapts a 3B-parameter MMDiT (Esser et al., 2024) architecture that was trained on a mixture of images, single-shot, and multi-shot videos at their native resolutions and durations. The model's full self-attention is expanded from per-shot scope to a scene-level context window of up to eight shots (roughly 8 seconds, 22k tokens each), using an interleaved 3D RoPE (Su et al., 2021) to give every shot distinct absolute coordinates while preserving the relative layout of text and video tokens. We initialize our model weights from pretrained LCT (Guo et al., 2025) and replace its attention module with our MoC, then fine-tune using the identical training scheme as LCT (Guo et al., 2025).

**Baselines.** We compare MoC with the base model LCT (Guo et al., 2025), which uses dense attention. For these experiments, we test on 8-shot sequences, where each shot is an 8-second 480p video with 12 FPS. This yields roughly 180k tokens per 64-second scene.

**Evaluation Metrics.** We follow prior work (Yin et al., 2025; Zhang & Agrawala, 2025) and evaluate on the popular VBench (Huang et al., 2024b;c) benchmark. Specifically, Subject Consistency and Background Consistency indicate how faithfully the primary subject and background from the input image are preserved throughout the video, Motion Smoothness evaluates the fluidity of movement (lack of jitter or abrupt transitions), and Dynamic Degree measures the extent of motion in the video (encouraging the generation of dynamic content rather than static scenes). We also report Aesthetic Quality and Image Quality to quantify each frame's visual appeal and technical quality. In addition, we report computational metrics such as sparsity, FLOPs, and inference speedup compared with Flash Attention (Dao et al., 2022; Dao, 2024).

**Quantitative Results.** Tab. 1 presents a quantitative comparison between our content-aligned Mixture of Contexts (MoC) model and dense attention baseline on minute-long multi-shot scenes. MoC exhibits clear computational advantages. By discarding 85% of the context, our approach achieves a $2.2\times$ speedup. Furthermore, it substantially enhances the performance of our model, particularly in terms of motion diversity, as evidenced by an increase in Dynamic-Degree from 0.46 to 0.56, while maintaining Motion-Smoothness. Although this increased motion budget leads to a slight reduction in appearance fidelity, all quality metrics remain high. Collectively, these results validate the core premise of our approach: learned, structure-aware sparsity reallocates computation from redundant frames to salient visual events, delivering significant efficiency gains without compromising (and in many cases improving) perceptual quality.

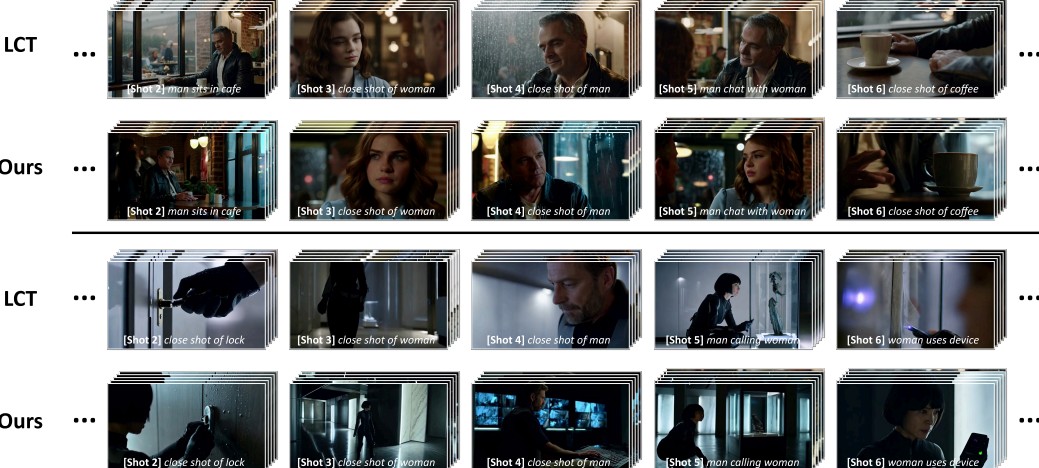

Figure 3: **Multi-shot video generation qualitative comparison.** Our results are visually indistinguishable from LCT (Guo et al., 2025), despite having pruned more than three-quarters of the attention calculation.

**Qualitative Results.** We present qualitative comparisons in Fig. 3. We argue that such a mean pooling operation is highly suitable for videos since pixels that lie close in space and neighboring frames tend to depict the same object or background region. After the DiT (Peebles & Xie, 2023)'s patch embedding, these tokens occupy a very narrow subspace: their first principal component often explains >90% of the local variance in practice. The arithmetic mean is exactly that first-component estimator for centered data, so a simple average already captures the dominant semantics of the whole chunk while discarding high-frequency noise. Zero-shot experiments support this claim – applying such a routing strategy directly to a pretrained video generation model, as will be shown in our supplementary material. Although the routing score in Eq. 3 is literally just a dot-product between the query and a mean-pooled key, it is not a fixed heuristic: the key vectors being averaged and the query vector doing the scoring are both produced by weights that are updated during training. Gradients flow through the mean-pool operation and the subsequent top-$k$ mask back to the projection matrices, allowing the model to learn how to shape each chunk's pooled key and each query in a way that best separates useful from irrelevant context. In practice, this makes the ostensibly "simple" mean + top-$k$ rule highly expressive without introducing extra routing parameters or computation, as the network continuously adapts its internal representations to exploit it.

**Qualitative Illustration of Coherence.** We provide a qualitative verification of long-term coherence in Fig. 4, demonstrating that Mixture of Contexts (MoC) robustly preserves consistency across diverse modalities and shot boundaries. The visualizations confirm that our learned dynamic sparse attention routing mechanism effectively maintains geometric background stability, fine-grained object details, and semantic alignment throughout the generation process. Furthermore, the model demonstrates strong multi-character/subject consistency, successfully distinguishing and preserving the unique identities of multiple subjects in dynamic scenes without feature mixing or identity drifting. MoC successfully retrieved these small details and highly abstracted semantic contents across hundreds even thousands of frames.

## 5 CONCLUSION

Adaptive Mixture of Contexts (MoC) demonstrates that learnable sparse attention routing can function as a powerful, data-driven memory retrieval engine. Our work is arguably the first to show that by scaling up training data with an efficient and learnable sparse routing mechanism, a model can develop a sophisticated method for long-term recall. This approach achieves minute-scale memory at a cost comparable to short-video generation. Critically, this capability emerges without explicit heuristics like 3D priors or Field-of-View (FoV) selection; the model learns entirely from data which historical context is salient. Because the routing is learned and the implementation is fast during

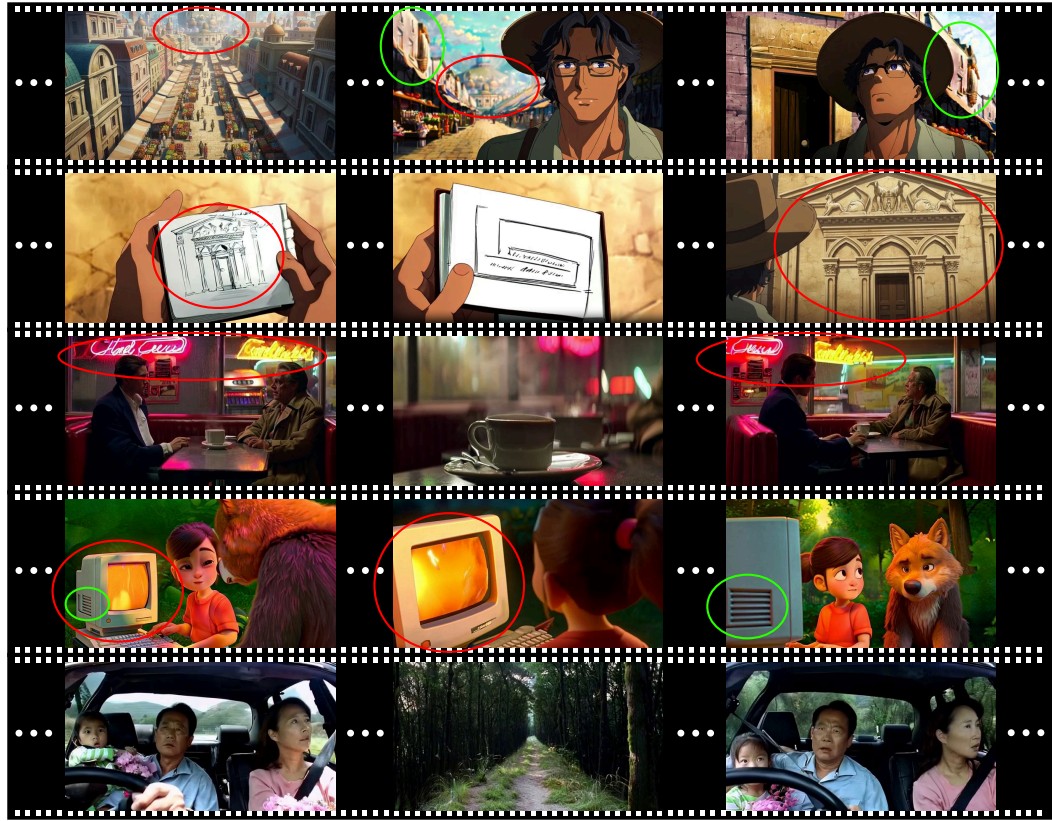

Figure 4: **Illustration of coherence achieved by Mixture of Contexts.** We highlight specific visual elements (indicated by red and green circles) that persist faithfully across shots, demonstrating the efficacy of MoC's retrieval mechanism. **Row 1:** Background landmarks (cityscape buildings) remain geometrically consistent despite camera movement. **Row 2:** Semantic consistency is maintained where a sketchbook drawing transitions into the corresponding physical architecture. **Row 3:** Spatial layout and background elements (neon signage) are preserved across reverse-angle cuts. **Row 4:** Fine-grained object identity is retained, such as the side vent structure (green) and screen (red) of the computer. **Row 5:** *Multi-character consistency* is achieved in a dynamic car interior, where the identities of the driver, passenger, and child remain distinct without feature mixing.

inference, MoC provides a blueprint for the next generation of scalable, controllable, and responsible long-video generative models. It proves that removing the quadratic attention bottleneck is not just an efficiency gain but a direct path to unlocking emergent, long-term memory in video generation.

**Limitation and Future Work.** So far, we have trained and tested on the identical setups as LCT (Guo et al., 2025). However, the ability of MoC to save computation on even longer sequences is yet to be explored. While our method already enables minute-scale context at near short-video cost, the current runtime relies on general-purpose variable-length attention and framework-level gathers. Given our FLOPs saving of $7\times$, substantial headroom for further speedups remains, which could be achieved with hardware–software co-design, e.g., block-sparse, chunk-aware var-len attention and more efficient customized CUDA/Triton kernels, fused routing+attention operators, persistent execution, and improved K/V layouts or quantization. We leave these extensions to future research.

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

## A  MEMORY COMPLEXITY ANALYSIS.

While sparse attention reduces computational complexity from $O(L^2)$ to roughly $O(k \cdot L)$, it introduces storage overhead for routing meta-data, specifically the mean-pooled keys, routing logits, and selection indices. However, this overhead is negligible in practice due to the coarse granularity of our chunking strategy. For a sequence length $L$ and chunk size $C$ (typically $C \in [10^3, 10^4]$), the number of chunks is $N \approx L/C$. Consequently, the memory required to store the representative mean-pooled keys scales as $O(N \cdot d)$, which is merely $1/C$ of the memory required for the full KV cache. Similarly, the routing logits matrix, which determines the top-$k$ selection, occupies $O(L \cdot N) = O(L^2/C)$ space; this represents a reduction by a factor of $C$ compared to a dense attention map. Crucially, our implementation minimizes peak memory usage by avoiding the materialization of intermediate expansions: we utilize `torch.segment_reduce` to compute pooled representations on-the-fly and encapsulate the sparse gather-scatter operations within a custom `torch.autograd.Function` (wrapping Flash-Attention kernels). This ensures that the memory footprint is dominated by the linear-complexity attention computation itself, with the routing overhead remaining a tiny fraction ($< 0.1\%$) of the total GPU memory budget.

## B  MoC IMPLEMENTATION BENCHMARK.

We benchmark our adaptive MoC's performance with full attention (implemented with Flash Attention 2 (Dao et al., 2022; Dao, 2024)) in Fig. 5, where our method stays near-linear in terms of FLOPs and latency with respect to the number of shots, or in other words, the sequence length

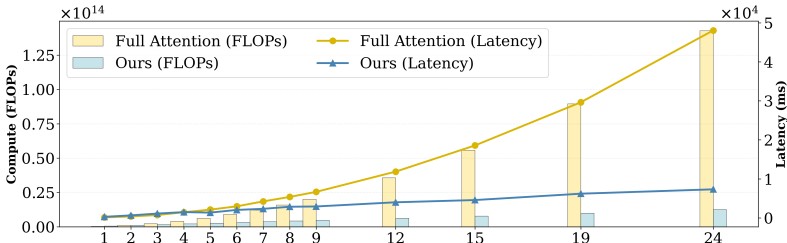

Figure 5: **Performance benchmark** of our content-aligned Mixture of Contexts implementation with full attention (implemented with Flash Attention 2 (Dao et al., 2022; Dao, 2024)). Our method stays near linear with respect to the shot number (xaxis, assuming 8 seconds, 12 FPS, roughly 23k tokens), or in other words, the sequence length $L$.

$L$. On top of sparsity, the key to this efficiency lies in three design decisions: (1) the use of on-the-fly `segment_reduce` pooling avoids materializing variable-length chunks in memory; (2) tokens are organized in head-major order to ensure coalesced memory access during gather operations; and (3) the entire routing + attention computation is wrapped in a single Flash Attention (Dao et al., 2022; Dao, 2024) var-len call, preserving kernel fusion and minimizing overhead.

## C  DATASET DETAILS.

Our main experiments are trained on a large-scale, scene-level multi-shot dataset curated from public narrative videos, following the data preparation protocol of LCT (Guo et al., 2025). Concretely, we collect long-form videos from publicly available sources across genres such as movies, TV series, and documentaries. Each raw video is first segmented into scenes using a standard scene boundary detector, and every scene is then further split into individual shots by shot-cut detection (PySceneDetect). A scene is thus represented as an ordered sequence of shots that share the same high-level semantics (characters, environment, storyline) but differ in local composition (framing, camera, micro-actions). For caption annotation, we use the multimodal model Gemini-1.5 as an automatic annotator and enforce a two-tier prompt structure similar to LCT (Guo et al., 2025). Each scene receives (i) a global caption that summarizes the shared context across all shots in the template "[Character] [Environment] [Story]", where characters are introduced as "Character [ID]: [Description]"; and (ii) a sequence of shot-level captions, one per shot. Shot-level captions describe the local action and camera/view for that shot, and crucially, they only refer to people via their global IDs ("Character 1", "Character 2", etc.) rather than ambiguous phrases like "the man/woman". This design makes character identity and environment explicit at the scene level while allowing shot captions to focus on fine-grained events and framing. This pipeline yields an authentic multi-shot dataset with approximately 500K annotated scenes, averaging about 5 shots per scene ($\approx 2.5M$ shot clips). To further enrich the data with smoothly evolving sequences that do not contain hard cuts, we additionally mine long single-shot videos that exhibit substantial temporal variation (e.g., moving cameras or actors). These videos are segmented into sub-shots based on detected event changes (rather than hard cuts) and treated as multi-shot scenes whose adjacent segments transition smoothly. This augmentation contributes roughly another 1M scene-level samples. In the authentic multi-shot subset (where shots come from real shot cuts), we prepend a special "[SHOT CUT]" token to the corresponding shot-level prompt to explicitly mark true transitions. Each training example therefore consists of a global scene caption, an ordered list of shot (or segment) clips, and aligned shot-level captions, plus an optional shot-cut marker, providing a transparent, reproducible scene-level dataset for long-context video generation.

## D  ZERO-SHOT EXPERIMENT

To isolate the benefit of the mean-pooled descriptor, independent of fine-tuning, we plug our MoC kernel directly into the pretrained dense model while freezing all weights. As shown in Fig. 6, despite never seeing sparse attention during training and high sparsity (>75%), the model maintains consistency reasonably. Because the descriptor is simply the arithmetic mean, it approximates the first principal component of each chunk, which is already well-aligned with dominant foreground/background patterns. This experiment highlights that the routing rule itself is data-adaptive, even without weight

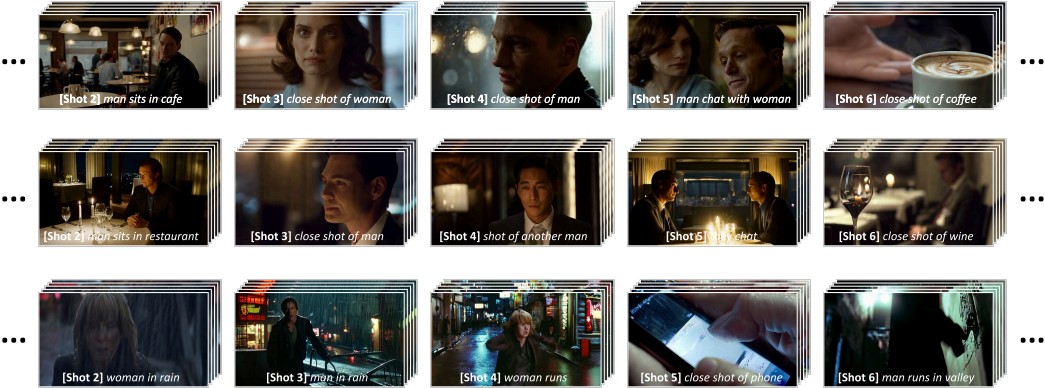

Figure 6: **Zero-shot sparsification.** We replace every dense attention block in a pretrained DiT with our Mixture of Contexts (>75% sparsity) without any fine-tuning. The model still preserves a certain amount of subject identity, background layout, and coarse motion, confirming that a simple mean-pooled chunk key already provides a usable retrieval signal even when the weights have never been exposed to sparse masks.

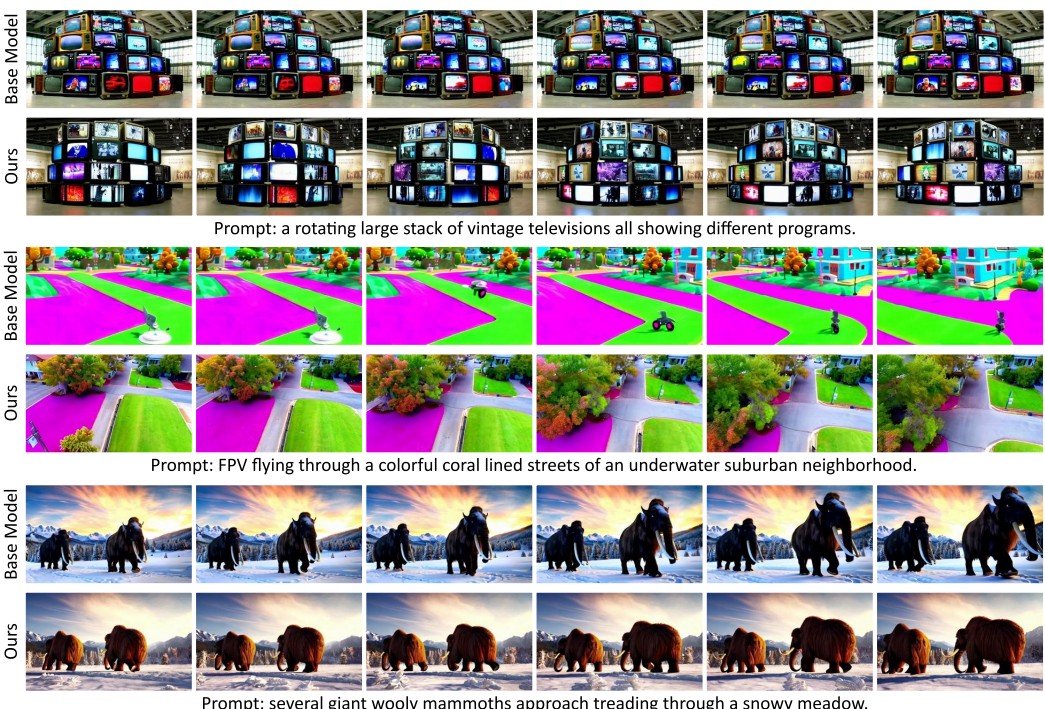

Figure 7: **Single-shot video generation qualitative comparison.** Our results are on par, if not better than, our base model despite aggressive sparsification.

updates, while learning can refine the query/key projections to make better use of it and increase its accuracy. These results validate our design choice: the parameter-free, mean-pooled descriptor is a strong, low-overhead signal that converts dense attention into a retrieval step, even in zero-shot settings. We note concurrent work such as VSA (Zhang et al., 2025d) has similar observations.

## E   SINGLE-SHOT SHORT VIDEO GENERATION

Our MoC specifically targets long, scene-level video generation with shot cuts, aiming to maintain context memory across long durations and various cuts. Nonetheless, we additionally supply experi-

| Method | Subject Consistency ↑ | Background Consistency ↑ | Motion Smoothness ↑ | Dynamic Degree ↑ | Aesthetic Quality ↑ | Image Quality ↑ | Sparsity ↑ | FLOPs ↓ |
|---|---|---|---|---|---|---|---|---|
| Base Model | 0.9380 | 0.9623 | 0.9816 | 0.6875 | 0.5200 | 0.6345 | 0% | $1.9 \times 10^{10}$ |
| **Ours** | **0.9398** | **0.9670** | **0.9851** | **0.7500** | **0.5547** | **0.6396** | **83%** | **$4.1 \times 10^{9}$** |

Table 2: **Single-shot video generation quantitative comparison.** We report VBench (Huang et al., 2024b) metrics and computation efficiency metrics. Our method is on par with or better than the base model for all VBench (Huang et al., 2024b) metrics despite aggressive sparsification (83%).

ments on short, shot-level text-to-video generation. We compare against the native 3B MMDiT (Esser et al., 2024) video generation model that is used as the very foundation of LCT (Guo et al., 2025) and our work. We test on 8-second videos with a resolution of $320 \times 192$ and 12 FPS, yielding roughly 6,300 tokens per video. Tab. 2 and Fig. 7 show the quantitative and qualitative results, respectively. For short single-shot videos (6k tokens), despite the aggressive sparsification, our method matches or surpasses the dense baseline across all VBench metrics. This demonstrates that directing computational resources toward the most relevant chunks not only reduces FLOPs but also enables the model to maintain character fidelity and scene coherence better. However, for such short sequences, the additional overhead from index gathering and pooling outweighs the computational savings, resulting in a slower end-to-end pipeline.

## F  TRAINING DETAILS

For our single-shot video generation model, we train jointly on images and videos. We use a chunk size of 256 and top-$k$=3, while enabling intra-chunk link and forced cross-modal link, where all chunks are forced to attend to themselves and the prompt tokens. We do not activate causality since we do not observe the pathologically closed-loop effect. For our multi-shot generation model, we train our model jointly on images, single-shot videos, and multi-shot videos using chunk size gradually decreasing from 10240, 5120, 2560 to 1280, and top-$k$=5, while enabling intra-shot link and forced cross-modal link, where each shot always performs self-attention, and each chunk attends to both local and global prompts. Both models are trained using a learning rate of $9e-5$, where the single-shot model is trained for 10k iterations and the multi-shot model is trained for 20k iterations.

## G  ABLATION STUDY

We systematically disentangle two design axes of our Mixture of Contexts: (1) effects of different chunk sizes and $k$ in our top-$k$ routing, and (2) the benefit of our forced links (cross-modal and intra-shot edges). For the former, we evaluate on single-shot video generation; for the latter, we focus on multi-shot video generation, where the forced links are more important. For the ablation study, we uniformly use 16 H100s and train 30k iterations for the single-shot experiments and 10k iterations for the multi-shot experiments, using a learning rate of $2e-5$.

| Chunk Size | $k$ | Subject Consistency ↑ | Background Consistency ↑ | Motion Smoothness ↑ | Dynamic Degree ↑ | Aesthetic Quality ↑ | Image Quality ↑ | Sparsity ↑ | FLOPs ↓ |
|---|---|---|---|---|---|---|---|---|---|
| 64 | 3 | 0.9868 | 0.9884 | 0.9928 | 0.3413 | 0.4964 | 0.6374 | 96% | $1.2 \times 10^{9}$ |
| 128 | 3 | 0.9909 | 0.9934 | 0.9937 | 0.2875 | 0.4634 | 0.6673 | 92% | $1.7 \times 10^{9}$ |
| 256 | 3 | 0.9916 | 0.9933 | 0.9938 | 0.4612 | 0.5283 | 0.6813 | 83% | $4.1 \times 10^{9}$ |
| 512 | 3 | 0.9649 | 0.9780 | 0.9873 | 0.5156 | 0.5275 | 0.6546 | 68% | $6.6 \times 10^{9}$ |
| 1024 | 3 | 0.9614 | 0.9736 | 0.9878 | 0.5938 | 0.5518 | 0.6471 | 35% | $1.3 \times 10^{10}$ |
| 256 | 1 | 0.9994 | 0.9995 | 0.9956 | 0.1313 | 0.3485 | 0.7421 | 92% | $2.1 \times 10^{9}$ |
| 256 | 2 | 0.9968 | 0.9966 | 0.9949 | 0.2781 | 0.4325 | 0.6940 | 88% | $3.1 \times 10^{9}$ |
| 256 | 3 | 0.9916 | 0.9933 | 0.9938 | 0.4612 | 0.5283 | 0.6813 | 83% | $4.1 \times 10^{9}$ |
| 256 | 4 | 0.9827 | 0.9863 | 0.9898 | 0.4531 | 0.5127 | 0.6276 | 80% | $5.2 \times 10^{9}$ |
| 256 | 5 | 0.9793 | 0.9848 | 0.9886 | 0.3594 | 0.5158 | 0.6456 | 76% | $6.2 \times 10^{9}$ |
| 256 | 6 | 0.9722 | 0.9805 | 0.9903 | 0.4219 | 0.5380 | 0.6629 | 72% | $7.2 \times 10^{9}$ |

Table 3: **Ablation study** on different chunk sizes and routing top-$k$.

**Chunk size and $k$.** Ablation results on different chunk sizes and $k$ are presented in Tab. 3. When we fix the number of retrieved chunks at $k$=3 and sweep the chunk length from 64 to 1024 tokens, we notice that tiny chunks (64,128) prune aggressively but harm motion, potentially because queries often lose access to far-context frames and are stuck with local optimums. We see similar trends with

fixing the chunk size at 256 and varying $k$ (each query also keeps its own chunk, so the effective fan-out is $(k + 1)$ This is a strong indication that a progressive approach that starts from larger chunks and larger $k$, then gradually switches to smaller chunks and smaller $k$ might be desired in order to achieve very aggressive sparsification.

| Force Intra-shot | Force Cross-modal | Context Drop In & Out | Subject Consistency ↑ | Background Consistency ↑ | Motion Smoothness ↑ | Dynamic Degree ↑ | Aesthetic Quality ↑ | Image Quality ↑ |
|---|---|---|---|---|---|---|---|---|
| ✗ | ✗ | ✗ | 0.8532 | 0.9391 | 0.9949 | 0.0000 | 0.2957 | 0.1552 |
| ✗ | ✓ | ✗ | 0.8305 | 0.9358 | 0.9952 | 0.0208 | 0.2934 | 0.1572 |
| ✓ | ✗ | ✗ | 0.9238 | 0.9446 | 0.9910 | 0.5729 | 0.5406 | 0.4472 |
| ✓ | ✓ | ✗ | 0.9323 | 0.9426 | 0.9890 | 0.4844 | 0.5442 | 0.5104 |
| ✓ | ✓ | ✓ | 0.9368 | 0.9579 | 0.9920 | 0.5469 | 0.5427 | 0.5061 |

Table 4: **Ablation study** on the effect of forced links.

**Force links and Context Drop In & Out.** Ablation on the effects of forced routing links is presented in Tab. 4. Experiments are conducted with a chunk size of 5120 and $k$=5. When the intro-shot link is not forced to be selected, we compensate the model to be able to select four additional chunks, which is roughly the number of tokens per shot. We notice that the training becomes extremely unstable when there are no forced intra-shot links to provide a sufficiently reasonable lower bound. Empirically, we find this to be highly relevant to the learning rate and batch size, while adding the intra-shot links makes the training much more stable. We also find that adding cross-modal links and Context Drop In & Out generally improves the overall performance of the model. This is consistent with the observations in Attention Sink (Xiao et al., 2024) and tricks typically used in sparse attention LLMs, where certain layers are designed as dense attention to enable better gradient flow, as in MoBA (Lu et al., 2025).

## H WAN-2.1-1.3B EXPERIMENT

| Method | Subject Consistency ↑ | Background Consistency ↑ | Motion Smoothness ↑ | Dynamic Degree ↑ | Aesthetic Quality ↑ | Image Quality ↑ | Sparsity ↑ |
|---|---|---|---|---|---|---|---|
| Dense Attention | 0.9512 | 0.9339 | **0.9869** | 0.4219 | 0.5154 | 0.5831 | 0% |
| **MoC (ours)** | **0.9549** | **0.9537** | 0.9833 | **0.6250** | **0.5204** | **0.6016** | **81%** |

Table 5: **Single-shot video generation quantitative comparison on Wan-2.1-1.3B.**

To demonstrate the generalization ability of MoA on general open-sourced backbones, we implemented and tested MoC on the Wan-2.1-1.3B model. We compare two settings: fine-tune the pretrained model using dense attention and our proposed Mixture-of-Attention. Since Wan-2.1-1.3B is not an MMDiT model but a regular DiT model, we apply MoC only on its self-attention modules using the same hyperparameters as our single-shot experiment. We train these two settings, each on 32 GPUs for 1 day (2000 iterations), using the Vchitect (Fan et al., 2025; Si et al., 2025) dataset at a resolution of 480p, with chunk size set at 1560 — number of tokens for a frame in Wan-2.1-1.3B. Results are presented in Tab. 5. We observe a similar trend to the aforementioned single-shot experiment, where sparsity is at least on par and often better than dense attention. This is solid proof of the generalization ability of MoC to other backbones, even without any model-wise adaptation of the MoC algorithm. We also find that our MoC performs reasonably well without many visible artifacts on Wan-2.1-1.3B, even without fine-tuning, as long as the sparsity does not become too low.

## I OUTER LOOP CONTEXT ROUTING

To further scale our approach to extremely long video sequences, we introduce an outer loop context routing mechanism in practice, which operates independently of the inner attention computation. Unlike the query-wise routing in Mixture of Contexts, which refines attention within selected chunks, the outer loop performs a preliminary selection of large-scale context chunks such as entire shot segments before any attention is computed. This pre-selection acts as a coarse filter, dynamically curating a subset of the global context to be fed into the subsequent Mixture of Contexts layers, thereby reducing the overall token pool and enabling linear scaling for sequences exceeding millions of tokens. Formally, given a flattened token stream partitioned into high-level chunks $\Psi = \{\Psi_1, \Psi_2, \ldots, \Psi_P\}$ where each $\Psi_j$ encompasses multiple lower-level chunks, the outer router computes

a global relevance score for each $\Psi_j$ relative to the current generation context. We employ the simple yet effective scorer again: a mean-pooled descriptor $\phi(\Psi_j) = \text{mean\_pool}(\boldsymbol{X}[\Psi_j])$, where $\boldsymbol{X}[\Psi_j]$ denotes the token features from all tokens in $\Psi_j$. For the current query block (e.g., the tokens of the shot being generated), we aggregate its token features into a single representative vector $x_g = \text{mean\_pool}(\boldsymbol{X}_g)$ and compute the similarity score as $\langle x_g, \phi(\Psi_j)\rangle$, where the top-$M$ large chunks are then selected $\Omega_g = \arg\max_{\Omega^* \subseteq \Psi, |\Omega^*|=M} \sum_{j \in \Omega^*} s_j$. The selected high-level chunks $\Omega_g$ are concatenated with mandatory elements (e.g., the global caption) to form a reduced context stream, which is then passed to the inner Mixture of Contexts for more fine-grained routing and sparser attention. This outer-inner hierarchy decouples coarse global retrieval from local refinement: the outer loop prunes redundant historical segments, while the inner loop focuses on precise token-level interactions within the curated subset. This is particularly helpful when dealing with extremely long contexts that scale beyond our training maximum length, as the outer loop compresses the effective context size to within the model's trained capacity, rendering the approach invariant to length extrapolation issues. Unlike dense attention mechanisms that suffer from positional embedding degradation (e.g., RoPE (Su et al., 2021) extrapolation problems leading to instability or performance drops beyond trained lengths), our hierarchical routing maintains stable positional encodings by operating on a curated, shorter subsequence, ensuring consistent performance even for arbitrarily long inputs without requiring specialized extrapolation techniques or retraining. The outer loop routing can effectively increase the number of shots we could generate by 2-3 times, under an autoregressive sampling strategy.

## J  SOCIAL IMPACT

Long-form video generators can democratize animation and documentary production, educational content, and simulation. Still, like all powerful generative models, they also lower the barrier for misinformation and non-consensual media synthesis. We advocate for a gated release, watermarking, and prompt filtering similar to current large-image and language models.

## K  THE USE OF LARGE LANGUAGE MODELS (LLMS)

We have used LLMs only to refine the writing of the paper, including rephrasing sentences and correcting grammatical mistakes. We hereby confirm this in accordance with the ICLR Author Guide.

