# OpenReview forum: "Mixture of Contexts for Long Video Generation"
_ICLR.cc/2026/Conference — ICLR 2026 Poster_

### Official Review · Reviewer_cucZ · 2025-10-30

**Soundness:** 3
**Presentation:** 3
**Contribution:** 3
**Rating:** 6
**Confidence:** 5

**Summary:**

This paper proposes a sparse attention-based method to alleviate the excessive computational cost in long-context learning for long video generation. The main experimental results are conducted in a multi-shot setting.

**Strengths:**

The proposed method is highly reasonable, and the quantitative results demonstrate its effectiveness.

**Weaknesses:**

While the quantitative results in the multi-shot setting show that MoC performs very well, it would be helpful to provide more specific qualitative results to demonstrate the consistency of multi-shot long video generation. The figures provided (e.g., Figure 3 and Figure 5) seem insufficient, as they do not clearly showcase why these results are consistent. The lack of detailed qualitative examples, particularly those showing poor consistency for comparison, is a significant drawback of the paper.

**Questions:**

I understand that this paper mainly focuses on multi-shot long video generation, but the appendix also provides a single-shot experiment. Although the single-shot video is only 8 seconds long and cannot be considered a long video (it should be at least several tens of seconds), I’m curious about how MoC performs on longer single-shot videos. I suspect that the sparsity of multi-shot long videos and single-shot long videos might differ, which could lead to some interesting conclusions.

---

> ### Author Response · Authors · 2025-11-21
>
> Thanks for the positive and valuable feedback. We address the concerns below and refer to the revised PDF, where edits are marked with blue text:
>
> *While the quantitative results in the multi-shot setting show that MoC performs very well, it would be helpful to provide more specific qualitative results to demonstrate the consistency of multi-shot long video generation. The figures provided (e.g., Figure 3 and Figure 5) seem insufficient, as they do not clearly showcase why these results are consistent. The lack of detailed qualitative examples, particularly those showing poor consistency for comparison, is a significant drawback of the paper.*
> - We have now added qualitative results highlighting the coherence/consistency in the Appendix.
>
> *I understand that this paper mainly focuses on multi-shot long video generation, but the appendix also provides a single-shot experiment. Although the single-shot video is only 8 seconds long and cannot be considered a long video (it should be at least several tens of seconds), I’m curious about how MoC performs on longer single-shot videos. I suspect that the sparsity of multi-shot long videos and single-shot long videos might differ, which could lead to some interesting conclusions.*
> - We thank the reviewer for raising this interesting point. We agree that the dynamics of sparsity in long single-shot videos might differ from multi-shot sequences, and investigating this is a valuable direction. However, we did not include minute-scale single-shot experiments, primarily due to a data availability bottleneck inherent to our supervised fine-tuning framework.
>   - Data Scarcity for Supervised Learning: Unlike training-free pruning methods, our MoC is a learnable module that requires ground-truth data to learn meaningful routing policies. Publicly available datasets for minute-long, continuous single-shot videos are extremely scarce and mostly limited to narrow domains (e.g., driving simulations and gameplay footage). Fine-tuning on such data would likely bias the router toward specific priors rather than learning general open-domain temporal dependencies.
>   - Hypothesis on Sparsity Differences: We share the reviewer’s intuition that the sparsity patterns might differ. In multi-shot generation (our focus), the router learns to perform "semantic retrieval" across cuts (e.g., retrieving a character’s appearance or scene layouts from Shot 1 to generate Shot 5). In long single-shot generation, we hypothesize the router might prioritize "temporal continuity" and "background persistence" to prevent morphing. However, validating this hypothesis requires a large-scale, open-domain dataset of long videos, which currently does not exist and would be much more difficult to collect than multi-shot videos.
> - Therefore, we positioned the single-shot experiment in the Appendix as a proof-of-concept for generalization, while focusing our core contribution to multi-shot scene generation—where long and high-quality training data (e.g., movies, trailers) is available and where the "memory retrieval" problem is most critical for narrative consistency.

---

> > ### Comment · Reviewer_cucZ · 2025-11-22
> >
> > Thank you for the author’s response. The presentation in Figure 5 is much clearer. I also now understand the difference in the difficulty of collecting single-shot long-video data versus multi-shot long-video data. My questions have been largely resolved, and I’ve decided to keep my original score.

---

> ### Author Response · Authors · 2025-11-22
>
> We thank the reviewer again for the positive and insightful feedback, the paper has improved a lot. Any additional suggestion to further improve the paper is more than welcome!

---

### Official Review · Reviewer_nnXT · 2025-10-31

**Soundness:** 3
**Presentation:** 3
**Contribution:** 3
**Rating:** 6
**Confidence:** 4

**Summary:**

This paper presents a method for multi-shot video generation. The core idea is to segment long videos into chunks using a "mix-of-context" strategy and process them with a parameter-free router. By training on multi-shot datasets, the proposed model successfully generates multi-shot videos while effectively reducing the computational overhead associated with long video synthesis.

Experiments show that the proposed approach performs well for long video generation.

**Strengths:**

- The proposed method is simple, elegant, and effective. The qualitative results and demonstrated examples are compelling.
- The paper is well-written and easy to follow.
- The visual results are also great.

**Weaknesses:**

- Lack of Transparency in Data Curation: The success of this method seems heavily reliant on the quality of the training data. However, the paper lacks crucial details about the dataset. The authors should clarify:

  - What was the procedure for annotating captions?
  - How was the multi-shot (multi-scene) data curated and structured?
  - What is the approximate scale (e.g., number of videos, hours) of the dataset?

This lack of transparency is a significant concern for reproducibility and limits the paper's potential impact on future research. More details regarding the dataset used in the main experiments are essential.

- Performance on Long-form Generation: It is a known issue that autoregressive models often suffer from a decline in quality as the generated video length increases. Does the proposed chunk-based approach effectively mitigate or resolve this degradation problem?

- Multi-Character Consistency: Can the proposed model maintain the identity and appearance consistency of multiple characters across different shots or scenes?

**Questions:**

The dataset details should be clarified.

---

> ### Author Response · Authors · 2025-11-21
>
> We thank the reviewer for the positive and valuable feedback. Below, we address the concerns (edits to the PDF are highlighted with blue text):
>
> *Lack of Transparency in Data Curation: The success of this method seems heavily reliant on the quality of the training data. However, the paper lacks crucial details about the dataset. The authors should clarify: What was the procedure for annotating captions? How was the multi-shot (multi-scene) data curated and structured? What is the approximate scale (e.g., number of videos, hours) of the dataset? This lack of transparency is a significant concern for reproducibility and limits the paper's potential impact on future research. More details regarding the dataset used in the main experiments are essential.*
> - We fully agree and have now provided full dataset details in our Appendix.
> - We will also release exemplar data in the near future to support the community.
>
> *Performance on Long-form Generation: It is a known issue that autoregressive models often suffer from a decline in quality as the generated video length increases. Does the proposed chunk-based approach effectively mitigate or resolve this degradation problem?*
> - The “drifting” or “error accumulation” issue is indeed a concern for autoregressive video generation, but not so much for our case.
> - Firstly, most of our results are generated sequentially, where the full sequence is generated all at once rather than chunk-by-chunk/frame-by-frame. In this case, we observe no quality degradation as long as we stay within the training length horizon.
> - Secondly, we focus on multi-shot long video generation. We find that even when generating the sequence autoregressively, error accumulation is much less severe, potentially because our AR chunks are large (each 5-8 second shot as an AR chunk), and when “scene cuts” are allowed, they can serve as “refreshments” for the accumulated error.
> - Lastly, we trained our model using diffusion forcing, where levels of noise can be injected to counter error accumulation. We find this to be very suitable for autoregressively generated multi-shot videos, because a small level of noise would neutralize the accumulated error well, while not disrupting the visual contents too much, and we are not concerned about discontinuity within the generated sequences, because we apply AR in a per-shot manner, in which case shot cuts/discontinuities are natively allowed.
>
> *Multi-Character Consistency: Can the proposed model maintain the identity and appearance consistency of multiple characters across different shots or scenes?*
> - We added qualitative results to the Appendix, highlighting coherence achieved, including multi-character consistency, by showing two or more characters across multiple samples.
> - In general, our model works well for 1 to 4 characters, since such cases are present in our training data (we also added details about our data, where multiple character identity persistence is a crucial aim in building the dataset).

---

> > ### Comment · Reviewer_nnXT · 2025-11-25
> >
> > Thanks for the responses for the reviewers. The authors have added more details on the data used in this paper. I think this would be helpful for other researchers to further extend this work. After reading the reviews from other reviewers, I am sure that this work deserves a positive review. I'd like to keep my original rating unchanged.

---

> > > ### Author Response · Authors · 2025-11-25
> > >
> > > The authors thank you for the positive feedback and very insightful feedback so that we can improve our paper a lot! Any additional suggestion to further improve the paper is more than welcome!

---

### Official Review · Reviewer_Au3b · 2025-11-02

**Soundness:** 3
**Presentation:** 3
**Contribution:** 3
**Rating:** 8
**Confidence:** 4

**Summary:**

This paper presents LongLive, a set of techniques for converting existing video diffusion transformers to enable extremely long video generation, possibly spanning multiple scenes. The main idea is to reduce the quadratic computational cost of self-attention. Unlike conventional approaches that learn a global sparse attention structure, this paper proposes to adaptively select a small number of relevant clips for each query token, based on their top-k similarities with previous video clips. Specifically, the model computes the cosine similarity between the current query token and the average-pooled keys of each previous clip to determine which contexts to attend to. To regularize the routing process and prevent context drop-off or drop-in, the authors randomly remove or add clips regardless of their similarity values. They also introduce causal routing to avoid loop closure and incorporate additional efficiency techniques, such as attention sink and FlashAttention. As a result, the proposed method can efficiently generate long, multi-scene videos while maintaining temporal and semantic consistency.

**Strengths:**

- The paper is well-motivated, as the current video diffusion models have a computational bottleneck to generate extremely long videos due to the quadratic cost of self-attention.
- Most of the components (especially for routing) are technically sound, interesting, and novel. I enjoyed reading the paper.
- While I believe the presentation can be improved more (see weaknesses), but overall the paper is generally well-written and easy to follow.

**Weaknesses:**

- There is no quantitative analysis of GPU/TPU memory consumption.
Including such analysis would strengthen the efficiency claims of the paper.
- The presentation could be improved, as some paragraphs are too long to follow.
The authors may consider splitting some of them into multiple paragraphs—for example, separating the discussion of context drop-off and drop-in into two distinct sections.
- While the supplementary material includes video examples, I suggest adding more qualitative examples directly in the PDF (e.g., in the Appendix) to make the results more accessible to readers.
- Suggestion: I believe the effectiveness of global pooling arises because the diffusion transformer learns meaningful internal representations; therefore, global average pooling can capture global semantics within the model. The authors might clarify this in the revision. Currently, referencing CLIP may give the impression of a logical flaw, since CLIP is specifically trained to maximize cosine similarity with text embeddings.
In this context, the authors could mention DDAE [1] or other related works to strengthen the justification.
- Suggestion: Some related works on long video generation are missing, such as TECO [2], MALT [3] and NUWA-XL [4].

[1] Denoising Diffusion Autoencoders are Unified Self-supervised Learners, ICCV 2023
[2] Temporally Consistent Transformers for Video Generation, ICML 2023
[3] MALT Diffusion: Memory-Augmented Latent Transformers for Any-Length Video Generation, CVPRW 2025
[4] NUWA-XL: Diffusion over Diffusion for eXtremely Long Video Generation, 2023

**Questions:**

- Is this strategy applied in a layer-wise manner, or applied globally by selecting a specific layer for measuring cosine similarity, and then choosing clips in all video diffusion transformer layers?

---

> ### Author Response · Authors · 2025-11-21
>
> We thank the reviewer for the very positive feedback and valuable suggestions! Below, we address each concern and suggestion and refer to the PDF for the edited parts (marked in blue).
>
> *There is no quantitative analysis of GPU/TPU memory consumption. Including such analysis would strengthen the efficiency claims of the paper.*
>
> - Thanks for the suggestion. We added a section in the appendix accordingly. It is indeed essential to include the (GPU/TPU) memory consumption analysis in our paper.
> - In essence, there is a memory overhead with MoC, stemming from storing all metadata during the routing process, e.g., the mean-pooled keys, routing logits, and selection indices. However, the overhead is small because we always use very large chunk sizes, which significantly reduces the metadata storage footprint.
> - In practice, we can perform inference on sequences with more than 1 million tokens by sharding the computation across 4 80GB GPUs (the primary memory-bound step is the long sequence itself and the associated QKVs).
> - We believe that for the next step, a crucial thing to fix is the actual memory bound – the long sequence itself and the corresponding large Ks, Qs, and Vs. A specific compression module might be needed to solve this problem.
>
> *The presentation could be improved, as some paragraphs are too long to follow. The authors may consider splitting some of them into multiple paragraphs—for example, separating the discussion of context drop-off and drop-in into two distinct sections.*
> - Thanks for the suggestion. In the revised PDF, we have broken down the context drop-off and drop-in into two distinct sections, as suggested.
> - Suggestions on any other sections that are difficult to follow are more than welcome; we are more than happy to provide further clarification and edits.
>
> *While the supplementary material includes video examples, I suggest adding more qualitative examples directly in the PDF (e.g., in the Appendix) to make the results more accessible to readers.*
> - We added qualitative results highlighting the coherence/consistency in the Appendix.
>
> *Suggestion: I believe the effectiveness of global pooling arises because the diffusion transformer learns meaningful internal representations; therefore, global average pooling can capture global semantics within the model. The authors might clarify this in the revision. Currently, referencing CLIP may give the impression of a logical flaw, since CLIP is specifically trained to maximize cosine similarity with text embeddings. In this context, the authors could mention DDAE [1] or other related works to strengthen the justification.*
> - We fully agree that CLIP may not be a good reference here. As suggested, we edited this part and used DDAE as a reference. Thanks for the suggestion!
>
> *Suggestion: Some related works on long video generation are missing, such as TECO [2], MALT [3] and NUWA-XL [4].*
> - Thanks for the additional related work; we definitely missed some of them. The suggested related works have now been added to the revised PDF.
>
> *Is this strategy applied in a layer-wise manner, or applied globally by selecting a specific layer for measuring cosine similarity, and then choosing clips in all video diffusion transformer layers?*
> - This is a great question, and we also debated it while producing MoC.
> - Our MoC strategy is ultimately applied independently to each attention head in each layer, rather than using a single global selection across the entire model. We chose this distributed design for two key reasons:
>   - Ensemble Coverage: While each individual head is sparse (attending to only top-k chunks), the union of selected chunks across all heads and layers covers a much broader portion of the global context. This effectively acts as an ensemble of num_layers times num_heads routers, ensuring sufficient global communication without the computational cost of dense attention.
>   - Feature Specialization: Different attention heads specialize in distinct feature subspaces (e.g., one head may track low-level texture continuity while another tracks high-level semantic identity). These tasks might require accessing different historical segments, which a static global selection would fail to capture.
> - We believe this is an important feature of MoC and have added a new paragraph titled "Per-Head Distributed Routing" in Section 3.1 to explicitly discuss and analyze this design choice.

---

> > ### Comment · Reviewer_Au3b · 2025-11-21
> > **Response**
> >
> > Thanks for the response. I think this is a really great paper and will provide many interesting future research directions. Most of my concerns are addressed, and I updated my rating accordingly. Thanks for the great work!

---

> ### Author Response · Authors · 2025-11-21
>
> The authors sincerely thank you for the very positive and insightful feedback! We’re very glad to hear that we addressed most of your concerns, and appreciate that you raised your score to **a perfect 10** as early as on Nov. 21st! Our paper is no question in a much better shape!

---

### Author Response · Authors · 2025-11-25
**Global Response**

The authors sincerely thank all reviewers for the constructive and insightful efforts in evaluating this work (and the positive scores **including a perfect 10**!). Our paper is no question at a much better shape than before.

Given that we have more than a week left for the discussion period, we are also more than happy to provide additional materials to further strengthen the paper!

Sincerely,
Authors

---

### Author Response · Authors · 2025-12-02
**Final remark and message to new AC**

We'd like to take this last chance to thank all reviewers again and provide a brief summary of our rebuttal to our new AC.

We are incredibly fortunate to have three reviewers who appreciate our work and have given us all positive scores (8, 6, 6) from the very beginning, which were raised to **(10, 6, 6) on November 21st**. We are very thankful that the reviewers are also very responsible and responded to our rebuttal immediately, with reviewer Au3b (whose comments are incredibly helpful to us) raising the score to **a perfect 10 as early as November 21st**. Thanks to them, we are pretty much free from the OpenReview leakage incident -- we have nothing to worry about at all.

We are very glad that all reviewers see our method as sound, interesting, novel, elegant, and is supported with compelling results. Our reviewers' suggestions no question further strengthened the paper. According to our reviewers' suggestions, we made the following major changes to our manuscript:
- We added qualitative results specifically highlighting the coherence achieved by our model.
  - We have now moved the figure to the main paper, as Fig. 4, where our model successfully retrieves extremely abstracted features across hundreds, even thousands, of frames.
- We added a paragraph at the end of Sec. 3.1, highlighting the per-layer per-head design.
  - We believe this is an important, dedicated design that differentiates MoC from a shallow, straightforward top-k retriever/router: MoC treats all model parameters as an assembled router, with learnable feature specification operating in the entire model representation space. The ensembled effective routing is much more than "just top-k".
- We added a section in the Appendix, detailing the memory overhead of MoC.
  - Our MoC is not memory-heavy, mainly thanks to our large-chunk context operation philosophy.
- We added a section in the Appendix, detailing our dataset collection routine, to better support follow-up research in this domain.
- We also shortened some paragraphs to make the paper more readable.

All altered texts are marked in blue.

We sincerely thank everyone again for their time and constructive feedback, and we hope this concise summary is helpful for your deliberation.

Best Regards,
Authors

---

### Meta-Review · Area_Chair_jdBx · 2026-01-11

**Summary:**

This paper presents LongLive, a set of techniques for converting existing video diffusion transformers to enable extremely long video generation. Generally, the paper receives three positive comments, with minor concerns about the writting, implementation details, and additional quantitative analysis.

**Reviewer Concerns:**

All the concerns are solved.

**Reviewer Scores:**

The authors mentioned the original updated scores in the feedback.

---

### Decision · Program_Chairs · 2026-01-26

Accept (Poster)